# Effects of Oral Amino Acid Supplementation on Physical Activity, Systemic Inflammation, and Quality of Life in Adult Patients with Cystic Fibrosis: A Single-Center, Randomized, Double-Blind, Placebo-Controlled Pilot Study

**DOI:** 10.3390/nu17071239

**Published:** 2025-04-02

**Authors:** Laura Petraglia, Paola Iacotucci, Lorenza Ferrillo, Serena Cabaro, Jolanda Somma, Francesca Lacava, Ilaria Amaranto, Silvia Crucito, Maria Perrotti, Pietro Formisano, Giuseppe Rengo, Dario Leosco, Vincenzo Carnovale

**Affiliations:** 1Department of Translational Medical Sciences, University of Naples “Federico II”, Via S. Pansini 5, 80131 Naples, NA, Italy; laura.petraglia@unina.it (L.P.); lorenza.ferrillo@unina.it (L.F.); serena.cabaro@unina.it (S.C.); jolanda.somma@unina.it (J.S.); francesca.lacava2@gmail.com (F.L.); sil.crucito@gmail.com (S.C.); maria.perrotti@unina.it (M.P.); pietro.formisano@unina.it (P.F.); giuseppe.rengo@unina.it (G.R.); vincenzo.carnovale@unina.it (V.C.); 2ANASTE Humanitas Foundation, 00192 Rome, RM, Italy; 3Department of Clinical Medicine and Surgery, University of Naples “Federico II”, 80131 Naples, NA, Italy; paola.iacotucci@unina.it; 4Emergency Department, AORN San Pio, 82100 Benevento, BN, Italy; amarantoilaria@gmail.com; 5Istituti Clinici Scientifici Maugeri IRCCS, Scientific Institute of Telese Terme, 82037 Telese Terme, BN, Italy

**Keywords:** cystic fibrosis, amino acid supplements, inflammation, interleukin 6

## Abstract

**Background/Objective:** Cystic Fibrosis (CF) is a common, life-threatening genetic disorder that leads to progressive lung function decline, respiratory failure, and premature death. Musculoskeletal complications, affecting both peripheral and respiratory muscles, are major concerns in CF patients. Inflammatory cytokines seem to be responsible for the activation of the molecular pathways involved in the imbalance between protein synthesis and catabolism, with consequent loss of muscle mass and function. This study aims to assess the effects of amino acid supplements on functional status, muscle mass and strength, inflammation, and quality of life in adult CF patients. **Methods:** We conducted a randomized, double-blind, placebo-controlled pilot trial with 60 adult CF patients, aged 18 or older. Participants were randomly assigned to receive either amino acid supplementation or a placebo for 4 weeks. Physical function tests and self-assessment questionnaires on quality of life, global health, and sleep status, as well as blood samples to measure pro-inflammatory cytokines, were performed at baseline and after the treatment period. **Results:** The amino acid supplementation group showed a significant improvement in self-perceived physical performance and health status. Interleukin-6 serum levels were significantly reduced in this group compared to those who received the placebo (*p* = 0.042). **Conclusions:** Amino acid supplementation in adult CF patients improves self-perception of health status and may reduce systemic inflammation, significantly decreasing serum levels of Interleukin-6. This suggests potential benefits for the overall well-being of CF patients and a reduction in their inflammatory status.

## 1. Introduction

Cystic Fibrosis (CF) is the most common genetic, multi-systemic, and life-threatening disease in the Caucasian population, with the average incidence ranging from 1:3000 to 1:6000 live births in North America and Western Europe [1].

CF is an autosomal recessive disease caused by mutations in the gene encoding the CF Transmembrane Conductance Regulator (CFTR), located on chromosome 7, and responsible for the efflux of chloride and bicarbonate anions in many organ systems. In the sweat glands, CF causes impaired chloride absorption and consequently elevated sweat chloride concentrations. The malfunction of CFTR in the airways leads to decreased chloride and bicarbonate secretion at the apical membrane, which causes thickened secretions and facilitates inflammation and bacterial infections of the airways, with a progressive decline in lung function up to respiratory failure and premature death [1,2]. Thickened secretions also cause pancreatic autodigestion with the development of pancreatic insufficiency, both exocrine and endocrine. Liver involvement ranges from steatosis to focal biliary fibrosis or to cirrhosis with portal hypertension requiring liver transplantation [2].

Relevant therapeutic advances have led to an increase in the median age of survival, which is about 61 years. However, the development of secondary musculoskeletal complications, which can involve peripheral muscles as well as respiratory muscles, still represents a major concern in the CF population [3]. Several factors, such as loss of muscle mass, poor nutritional status, systemic inflammation, corticosteroid use, and low physical activity levels, may contribute to the development of muscle dysfunction in adults with CF [3]. Furthermore, the caloric intake in patients with chronic respiratory diseases is often inadequate compared to their energy expenditure and, in most CF patients, the insufficiency of the exocrine pancreas leads to malabsorption [4,5]. It is of note that the close association between loss of strength and muscle mass and elevated serum levels of inflammatory proteins such as C-Reactive Protein (CRP), Tumor Necrosis Factor-α (TNF-α), Interleukin-6 (IL-6), and Interleukin-8 (IL-8) is described. The inflammatory cytokines seem to be responsible for the activation of the molecular pathways involved in the imbalance between protein synthesis and catabolism, with consequent loss of muscle mass and function [6,7]. Chronic systemic inflammation is reported in patients with chronic respiratory diseases, such as CF patients, with an increase in circulating leukocytes and elevated serum levels of CRP, Fibrinogen, TNF-α, IL-6, and 8 [7,8,9]. This condition can contribute to the development of muscle dysfunction [10]. Furthermore, inflammation can promote anorexia and malnutrition that characterize CF, thus establishing a vicious circle [11,12]. Given the association between sarcopenia and poor prognosis, it is important to identify therapeutic interventions aimed at maintaining adequate levels of muscle function and mass. Several pieces of evidence suggest that nutritional supplements can be used in addition to a balanced diet to enhance its effects [4,5]. Therefore, the present study aims to evaluate the effects of aminoacidic supplements on the general functional status, muscle mass and strength, inflammatory status, and quality of life of a population of adult CF patients.

## 2. Materials and Methods

### 2.1. Trial Design and Study Population

We designed a randomized, double-blind, single-center, placebo-controlled, parallel-group pilot trial involving patients of 18 years or older with CF. The trial was conducted at the Regional Cystic Fibrosis Adult Center of the University of Naples Federico II, between April and November 2021. We enrolled sixty patients with a predicted forced expiratory volume in 1 s (FEV_1_) between ≥40% and ≤80%, and randomly assigned them in a 1:1 ratio to receive therapy with aminoacidic supplementation or a placebo for 4 weeks. Randomization was stratified according to age, sex, and the percentage of the FEV_1_ at screening visits. Therapy with amino acid nutritional supplementation consisted of 2 sachets containing 20.6 kcal, 4 g amino acids (including essential amino acids), 0.15 g fat, 0.25 g carbohydrate, and vitamins B1 and B6 (the detailed nutritional composition is presented in Table 1). The placebo and supplementation formulas differ only in amino acid content, while maintaining the same amount of fat and carbohydrates. Following international guidelines [13], the protein intake for all CF patients enrolled in the study was 1.2–1.5 g/kg (almost 20% more than the general population). Given the similar dietetic regimen for all CF patients, the extra content of EAA per kilogram was ≈12.5%, equal to 0.15 g/kg, in the group with aminoacidic supplementation.

The exclusion criteria were carefully decided and were represented by: a history of any comorbidity that, in the opinion of the investigator, could confound the results of the study or pose an additional risk in administering the aminoacidic supplementation; abnormal laboratory values at screening visits such as a glomerular filtration rate of ≤50 mL/min/1.73 m; an acute upper or lower respiratory infection, pulmonary exacerbation, or changes in therapy (including antibiotics and CFTR modulator therapies) for pulmonary disease in the 8 weeks before the start of the trial; colonization with organisms associated with a more rapid decline in pulmonary status (e.g., Burkholderia cenocepacia and Mycobacterium abscessus); history of solid organ or hematological transplantation; history of alcohol or drug abuse in the past year; pregnancy and breastfeeding; and therapy with other amino acid supplements. This study included a screening period (4 weeks) (Appendix A: Table A1), a treatment period (4 weeks), and a follow-up visit (Appendix A: Table A2).

The trial protocol and informed consent forms were approved by the independent local ethics committee of the University of Naples “Federico II” Italy (N. 27/21, 15 March 2021). All enrolled patients provided written and informed consent.

### 2.2. Physical Assessments

All patients underwent a complete clinical examination, blood tests, and spirometry for the determination of FEV_1_, Forced Vital Capacity (FVC), the FEV_1_/FVC (ratio), and Forced Expiratory Flow (FEF 25–75%), at baseline and after four weeks of treatment. A pregnancy test was performed for women patients.

Before starting treatment and after four weeks, the patients underwent the physical function tests described below. Due to the known clinical complexity of adult CF patients, whose condition involves frailty, we used physical performance tests validated for elderly patients.

#### 2.2.1. The 30 s Sit-to-Stand Test

The 30-s Sit-to-Stand Test, also known as the 30 s chair stand test (30CST), was used to test leg strength and endurance. The patients were encouraged to complete as many full stands as possible within 30 s, fully sitting between each stand [14]. The test was administered using a folding chair without arms.

#### 2.2.2. Stair Climb Power Test

The Stair Climb Power Test (SCPT) was recently proposed as a simple and safe test associated with measures of lower-limb muscle strength, power, and functional performance. Because the SCPT does not require additional equipment, it could be a reasonable alternative to more-sophisticated tests for measuring lower-limb muscle impairments in people with CF [15]. Patients were timed as they climbed a 10-step flight of stairs as quickly as possible, and the heart rate and oxygen saturation were monitored during exercise.

#### 2.2.3. Vertical Jump Height

This test measures the explosive power of the lower limbs through the maximum height reached with a jump. The patient stood next to a wall and, keeping the feet flat on the ground, marked or recorded the point of the fingertips. This is called the standing reach height. The patient put chalk on their fingertips to mark the wall at the height of their jump, stood away from the wall, and jumped vertically using both arms and legs to assist in projecting the body upwards, attempting to touch the wall as high as possible. The difference in distance between the standing reach height and the jump height represents the score. The best of three attempts was recorded [16].

#### 2.2.4. Triple Hop Distance

For the Triple Hop Distance, a tape measure was placed along the floor. The patients, starting at one end of the tape, had to hop 3 times consecutively on their dominant leg, trying to cover as much distance as possible. The distance hopped is recorded in centimeters [17].

### 2.3. Quality of Life Assessments

All patients, at the beginning of treatment and after 4 weeks, were administered three questionnaires, two of which were designed specifically for CF patients.

#### 2.3.1. Cystic Fibrosis Questionnaire–Revised (CFQ-R)

CFQR is a disease-specific health-related quality of life (HRQOL) measure for children, adolescents, and adults with CF. It is an instrument designed to measure impact on overall health, daily life, perceived well-being, and symptoms [18].

#### 2.3.2. The 12-Item Short Form Survey (SF-12) (Version 1.0)

SF-12 v2 is a general health questionnaire, providing information about physical and mental health. Two summary scores are reported from the SF-12: a mental component score (MCS-12) and a physical component score (PCS-12). The scores may be reported as Z-scores (difference compared to the population mean, measured in standard deviations, SD). The United States population averages for PCS-12 and MCS-12 are both 50 points, and SD is 10 points. So, each increment of 10 points above or below 50 corresponds to one standard deviation away from the average.

#### 2.3.3. Pittsburgh Sleep Quality Index (PSQI)

PSQI is a self-administered questionnaire which evaluates sleep quality and disturbances over a one-month time interval. Nineteen individual items measure seven component scores for sleep quality: subjective sleep quality, sleep latency, sleep duration, habitual sleep efficiency, sleep disturbances, use of sleeping medication, and daytime dysfunction. The sum of these seven component scores produces a global score.

### 2.4. Systemic Inflammation

At day 1 and at week 4, a peripheral venous blood sample was taken from all patients, placed into EDTA tubes, then centrifuged for 20 min at 1500 rpm and for 10 min at 4 °C. Plasma was stored in aliquots at −80 °C for ELISA assays. Plasma levels of adipokines and cytokines/chemokines (IL-6, IL-8, IL-10, TNF-α, and Monocyte Chemoattractant Protein-1–MCP-1) were determined using ELISA. Serum hsCRP was also evaluated.

### 2.5. Statistical Analysis

Means and standard deviations for continuous outcomes were reported. The normality of the data was evaluated using the Shapiro–Wilk and Kolmogorov–Smirnov tests; if the *p*-value was greater than 0.05, then the data were found to be non-normal and the paired *t*-test was replaced by the Wilcoxon signed-rank test, while the independent samples *t*-test was replaced by the Wilcoxon rank-sum test.

The follow-up change from the baseline within the study group was analyzed using a *t*-test for repeated measures (paired samples *t*-test). Instead, to compare the differences between the two study groups in terms of the follow-up change from the baseline, a Generalized Linear Model (GLM) was used. The GLM includes the treatment group as factor and the baseline value as covariate. No replacement method was used for missing data. For every test, *p* ≤ 0.05 was used as the significance threshold. All analyses were performed using SPSS (Statistical Package for Social Science for Windows, Version 21.0, IBM Corp., Armonk, NY, USA).

## 3. Results

Of the 60 patients identified as the study population, complete data were obtained for 55 subjects at the baseline visit; however, due to some dropouts, follow-ups were correctly completed for only 48 patients. No side effects were observed in either treatment group during the study.

The mean age of the study population was 32.8 ± 11.56 years and 54.5% of patients were females. The mean body mass index (BMI) was 24.37 ± 3.78 Kg/m^2^. At baseline, there were no statistically significant differences between the two groups in treatment with aminoacidic supplementation or the placebo in terms of respiratory function assessed via spirometry (FEV1, FVC, FEV1/FVC, and FEF), physical function tests, and self-assessment questionnaires on quality of life and physical and mental health status (Table 2 and Table 3).

As regards the genetic mutations responsible for the pathology, DELTAF508/W1282X was found to be the most represented genotype, equally distributed between the two groups, with 15 patients in the placebo group and 18 in the treatment group. The second most frequent mutation was N1303K, prevalent in the group of patients treated with amino acid supplementation (6 vs. 1). A detailed descriptive table (Appendix A: Table A3) was provided to outline all genotypes of the study patients.

At the follow-up visits, after four weeks of treatment, non-significant differences were maintained in the physical function tests. However, an improvement was observed in the self-perception of physical performance, assessed through the physical domain of the SF-12 questionnaire, in favor of the aminoacidic supplementation group (*p*-value = 0.045). No differences were found for the SF-12 mental score or the quality of sleep, investigated using the Pittsburgh questionnaire. Concerning the CFQ-R, one quality of life domain and one symptom domain showed a significant difference (*p*-value < 0.05) between the two groups: treatment (*p*-value = 0.044, placebo arm mean = 6.97 vs. aminoacidic supplementation arm mean = −1.02) and digestion (*p*-value = 0.006, placebo arm mean = −9.16 vs. aminoacidic supplementation arm mean = 7.08).

Table 4 illustrates all changes from the baseline from the respiratory and physical function tests and self-evaluating questionnaires.

Although serum CRP levels increased by more than 100% (+107.1%) in the placebo-treated group compared to a reduction of −8.1% in the amino acid supplementation group, this difference was not statistically significant (*p*-value = 0.282).

IL-6, IL-8, IL-10, TNF-α, and MCP-1 were detected in serum samples. At follow-up, IL-6 levels were significantly reduced in the amino acid supplementation group compared to those in the placebo (*p* = 0.042) (Figure 1). However, the mean levels of other pro-inflammatory cytokines, with the exception of MCP-1, show a tendency towards a reduction in favor of the amino acid supplementation treatment compared to the placebo group. Notably, IL-8 approaches statistical significance (*p* = 0.099) (Table 5).

## 4. Discussion

Several pieces of evidence suggest a correlation between chronic respiratory diseases and the reduction of strength and muscle mass, mainly mediated by chronic inflammatory processes, low levels of physical activity, and poor nutritional status [6,7,8,9,10,11,12].

The primary objective of our randomized, double-blind and placebo-controlled pilot study was to evaluate the effect of aminoacidic supplementation on systemic inflammation, physical performance, and quality of life in a population of adult patients with CF. In the present study, nutritional support induced: (1) a significant reduction in IL-6 serum levels; (2) a significant improvement in physical health status perception, assessed through the SF-12 questionnaire; and (3) a significant reduction in the perceived treatment burden and a significant improvement in digestive symptoms, evaluated with the CFQ-R questionnaire.

CF is characterized by severe lung inflammation [19,20,21]. Airway epithelial cells, macrophages, and neutrophils can produce pro-inflammatory cytokines. Elevated concentrations of IL-6, IL-1, IL-8, and TNFα have been found in the sputum and bronchoalveolar lavage fluid of CF patients. These cytokines are responsible for forming an excessive neutrophilic infiltrate, which perpetuates an endless loop, promoting the production of other inflammatory mediators [20,21]. To date, no data are available on the effects of supplementation with a mix of essential amino acids on serum levels of pro-inflammatory cytokines in CF patients. The present study demonstrates a significant reduction in serum IL-6 levels following amino acid supplementation, thus suggesting a possible therapeutic intervention capable of decreasing the inflammatory cascade in CF. IL-6 is one of the main cytokines, responsible for triggering and perpetuating the inflammatory cascade [22]. Its role in the development and progression of numerous inflammatory and chronic degenerative diseases, such as in CF, is widely recognized [21,22,23]. Dietary interventions aimed at improving muscle mass representation in patients with chronic respiratory diseases may help interrupt the negative bidirectional mechanisms between muscle mass loss and systemic inflammation.

The serum concentration of other inflammatory factors analyzed in the present study shows a clear, although non-significant, trend towards a reduction in the group of patients treated with aminoacidic supplementation, in particular in IL-8, a cytokine whose role has been extensively studied in CF patients [21,24,25]. It is reasonable to hypothesize that the lack of significant differences between the two treatment arms could be ascribed to the low sample size.

As described in the results section, no statistically significant differences were found between treated and untreated patients in serum hsCRP levels. This result could be due to the lower sensitivity of this inflammatory marker in patients without pulmonary exacerbations. Several pieces of evidence demonstrate that CRP is a useful marker of acute infection and exacerbation in patients with CF [26,27]. In our study, we excluded patients with acute upper or lower respiratory tract infections or pulmonary exacerbations.

It is widely recognized that CF is a chronic, multi-systemic disease with multiple comorbidities that greatly affects quality of life [28]. The often coexisting respiratory and gastrointestinal symptoms, and the resulting physical limitations associated with the loss of muscle mass and function [2,3,4], lead to important psychological consequences and an unavoidable reduction in quality of life. The results of this pilot study show a significant improvement in some domains of the quality of life questionnaires. In particular, after 4 weeks of therapy with aminoacidic supplements, the domains related to digestive function and treatment burden, explored through the CFQ-R, were significantly improved. This questionnaire is a self-reported tool specifically designed to assess the health status of individuals with CF [18]. It evaluates various domains of health, including physical, respiratory, digestive, emotional, and social functioning, as well as vitality, described as the level of energy and fatigue experienced by the patient. Treatment burden refers to the physical, emotional, and logistical challenges that CF patients face in managing their treatment regimens. This includes the time spent on treatments, the difficulty of adhering to complex medication schedules, and the emotional toll of managing a chronic illness. Questions dedicated to treatment burden help assess the overall impact of therapies and disease on patients’ quality of life. Focusing on gastrointestinal symptoms, the CFQ-R includes specific questions dedicated to the frequency of abdominal pain, bloating, and bowel movement issues. These questions aim to capture how CF-related digestive problems affect the patient’s daily life and overall health status. In this regard, it is important to note that CF patients exhibit alterations in the intestinal microbiota, probably partly due to the mucosal inflammation disease [29]. Furthermore, it should also be underlined that most of the study population presented genotypes associated with more severe disease manifestations, with significant respiratory and digestive involvement. The DELTAF508/W1282X mutation, presented in 33 patients and equally distributed between the two study groups, includes the DELTAF508 mutation, the most common CF mutation, paired with the W1282X mutation, which is a nonsense mutation. This combination typically leads to a more severe phenotype with significant pulmonary and gastrointestinal complications [30]. In our study, a significant improvement in general physical condition and self-perception of well-being is also demonstrated by the results of the physical component score of the SF-12 questionnaire. The SF-12 is a tool used to assess the physical and mental component of health-related quality of life. Physical score measures various aspects of physical function, such as the ability to perform daily activities and the presence of pain or fatigue, and is employed to evaluate the impact of the disease on physical health. However, despite the results obtained from the self-assessment questionnaires, there were no significant differences between the physical function tests of the amino acid supplementation and placebo groups. In this regard, it should be underlined that the nutritional supplement contained 30% leucine enrichment, as illustrated in Table 1. Several pieces of evidence show the positive effects of leucine supplementation on strength and muscle mass in sarcopenic patients [31,32]. Adult CF patients, especially with severe manifestations of the disease, effectively represent a model of frailty and sarcopenia [3,6]. Therefore, the absence of significance in the physical tests could have several potential explanations. It is likely that the low sample size of our study population hindered detecting a significant improvement in functioning in the active treatment group. Furthermore, the duration of treatment, of only 4 weeks, was probably too short to significantly impact physical performance and status in CF patients. Finally, the assessment of physical function through validated tests for elderly patients, due to the aforementioned conditions of frailty and sarcopenia, together with the short duration of treatment, could have negatively affected the results of the physical function tests. However, due to the complexity of the pathology and the important psychological repercussions it entails, the improvement in quality of life should be taken into great consideration.

## 5. Conclusions

CF unfortunately represents a unique combination of chronic infections and dysfunctional immune responses that, in turn, contribute to inflammation and lung destruction. The success of modulatory therapies in improving lung function and life expectancy does not reduce the need to counteract the pro-inflammatory state that characterizes this disease and to improve the quality of life and physical and muscular performance of CF patients. Amino acid supplementation could represent a valid therapeutic support to achieve these goals. Further larger-scale studies will be necessary to confirm our results.

## 6. Study Limitations

This study was designed as a randomized, double-blind, single-center, placebo-controlled, parallel-group pilot study. The main limitations are certainly the small number of patients, representative of a single center, and the treatment duration of only 4 weeks. Furthermore, it was not possible to collect information on lean and body mass at baseline and follow-up for all the study population. It will be necessary to expand the study population and hypothesize a 3-month treatment with aminoacidic supplementation to improve and confirm these promising results on systemic inflammation, quality of life, and physical performance in CF.

## Figures and Tables

**Figure 1 nutrients-17-01239-f001:**
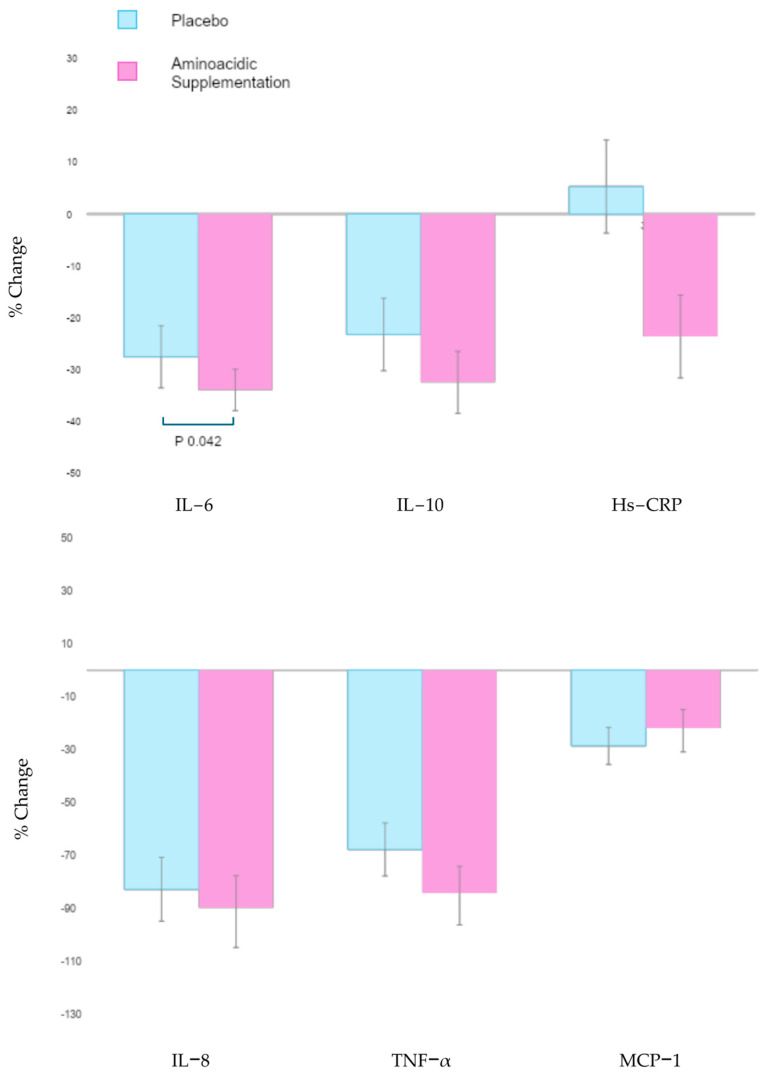
Changes from baseline of Interleukins and hs-CRP showing an IL-6 reduction greater than 40% in aminoacidic supplementation group (*p* = 0.042). IL: Interleukin; MCP-1: Monocyte Chemoattractant Protein-1; hs-CRP: high sensitivity C-Reactive Protein; TNF-α: Tumor Necrosis Factor-α.

**Table 1 nutrients-17-01239-t001:** Nutritional composition of amino acid supplementation (per 1 sachet).

Energy Value	20.6 Kcal
Fats	0.15 g
Carbohydrates	0.25 g
Protein	0 g
L-Leucine	1250 mg
L-Lysine	650 mg
L-Isoleucine	625 mg
L-Valine	625 mg
L-Threonine	350 mg
L-Cystine	150 mg
L-Histidine	150 mg
L-Phenylalanine	100 mg
L-Methionine	50 mg
L-Tyrosine	30 mg
L-Tryptophan	20 mg
Vitamin B6	0.15 mg
Vitamin B1	0.15 mg

**Table 2 nutrients-17-01239-t002:** Population characteristics at baseline evaluation.

	Placebo				Aminoacidic Supplementation			
	N	Median	Min	Max	N	Median	Min	Max
Age (years)	26	31	18	58	27	28	20	62
BMI (kg/m^2^)	26	24.7	18	32	26	23	19	38
HR	24	83	60	126	26	87	50	130
RR	20	23	16	30	21	24	13	32
FEV_1_	26	2.1	0.6	5.4	29	2.1	0.8	4.3
FVC	25	3.3	1.5	6.1	29	3.1	1.6	5.2
FEV_1_/FVC	25	0.7	0.5	0.9	29	0.7	0.4	0.9
30 s Sit-to-Stand Test	25	14	11.	30	29	15	4.0	26
Vertical Jump Height	23	27	8	43	24	2	6.0	42
Triple Hop Distance	26	271	120	560	29	270	90	510

BMI: body mass index; HR: heart rate; RR: respiratory rate; FEV_1_: predicted forced expiratory volume in 1 s; FVC: Forced Vital Capacity.

**Table 3 nutrients-17-01239-t003:** Questionnaire scores at baseline evaluation.

	Placebo				Aminoacidic Supplementation			
	N	Median	Min	Max	N	Median	Min	Max
SF-12 Physical Score	25	48.3	31	59.7	28	50.4	20	58
SF-12 Mental Score	25	49.1	28.3	67.2	28	53.2	31	62.5
PSQI	26	5	1	16	29	4	0	14
CFQ-R Physical	26	60.4	13	100	29	66.7	4	100
CFQ-R Vitality	26	58.3	25	100	29	66.7	17	100
CFQ-R Emotion	26	80	7	100	29	86.7	40	100
CFQ-R Eat	26	100	67	100	29	100	44	100
CFQ-R Treat	26	44.4	11	89	29	55.6	22	100
CFQ-R Health	26	66.7	22	100	29	77.8	33	100
CFQ-R Social	26	66.7	28	100	29	72.2	28	100
CFQ-R Body	26	77.8	33	100	29	88.9	33	100
CFQ-R Role	26	79.2	33	100	29	91.7	42	100
CFQ-R Weight	26	100	0	100	29	100	33	100
CFQ-R Respiratory	26	66.7	17	100	29	77.8	33	100
CFQ-R Digestive	25	77.8	22	100	29	77.8	33	100

SF-12: Short Form 12 Health; PSQI: Pittsburgh Sleep Quality Index; CFQ-R: Cystic Fibrosis Questionnaire–Revised.

**Table 4 nutrients-17-01239-t004:** Changes from baseline in respiratory and physical function tests and self-evaluating questionnaires.

	Placebo (N = 26)	Aminoacidic Supplementation (N = 29)	*p* Value
FEV_1_ (mean, SD)	2.3 (1.2)	2.2 (0.9)	0.706
FVC (mean, SD)	3.3 (1.2)	3.1 (0.9)	0.651
FEV_1_/FVC (mean, SD)	−0.01(0.1)	−0.02 (0.1)	0.859
30 s Sit-to-Stand Test (mean, SD)	1.4 (4.5)	3 (3.1)	0.055
Stair Climb Power Test (FC) (mean, SD)	0.5 (19.1)	4.3 (23)	0.379
Vertical Jump Height (mean, SD)	0.3 (6.2)	1.3 (10.2)	0.496
Triple Hop Distance (mean, SD)	18.3 (71)	13.5 (79.2)	0.857
SF-12 Physical Score (mean, SD)	−2.8 (7.3)	2 (9.9)	**0.045**
SF-12 Mental Score (mean, SD)	−1.2 (7.4)	0.8 (8.7)	0.144
PSQI (mean, SD)	−1.4 (3.5)	0.2 (1.9)	0.062
CFQ-R Physical (mean, SD)	−4 (14.5)	1.2 (17.2)	0.297
CFQ-R Vitality (mean, SD)	−3.5 (13.2)	−0.3 (12.5)	0.371
CFQ-R Emotion (mean, SD)	−2.6 (23.1)	1.5 (17.2)	0.921
CFQ-R Eat (mean, SD)	−1.4 (14.4)	0.0 (8.4)	0.394
CFQ-R Treat (mean, SD)	7 (17)	−1 (13.3)	**0.044**
CFQ-R Health (mean, SD)	−4.6 (15.7)	−2 (10.1)	0.238
CFQ-R Social (mean, SD)	0.5 (12.8)	3.3 (8.9)	0.346
CFQ-R Body (mean, SD)	−2.3 (16.7)	0.0 (9.7)	0.981
CFQ-R Role (mean, SD)	0.0 (14.1)	0.01 (9.6)	0.982
CFQ-R Weight (mean, SD)	−2.8 (25.8)	0.0 (20.6)	0.691
CFQ-R Respiratory (mean, SD)	−0.01 (21.5)	−2.5 (14)	0.573
CFQ-R Digestive (mean, SD)	−9.2 (21.4)	7.1 (16.7)	**0.006**

FEV_1_: predicted forced expiratory volume in 1 s; FVC: Forced Vital Capacity; SF-12: Short Form 12 Health; PSQI: Pittsburgh Sleep Quality Index; CFQ-R: Cystic Fibrosis Questionnaire–Revised. Bold in the *p*-value is necessary to stress the significant results.

**Table 5 nutrients-17-01239-t005:** Changes from baseline in Interleukin serum levels.

	Placebo (N = 26)	Aminoacidic Supplementation (N = 29)	*p* Value
IL-6 (pg/mL)			
N	23	22	
Baseline, mean (SD)	6.70 (9.5)	6.27 (5.8)	
End of study, mean (SD)	4.85 (8.7)	4.14 (3.8)	
Changes (%) from baseline, mean (SD)	−27.6 (6)	−34.0 (4)	**0.042**
IL-8 (pg/mL)			
N	22	22	
Baseline, mean (SD)	101.35 (150.5)	149.12 (217.1)	
End of study, mean (SD)	17.51 (17.2)	15.40 (9.4)	
Changes (%) from baseline, mean (SD)	−82.72 (15)	−89.67 (12)	0.099
IL-10 (pg/mL)			
N	22	20	
Baseline, mean (SD)	1.89 (1.7)	2.09 (1.5)	
End of study, mean (SD)	1.45 (1.5)	1.41 (1.1)	
Changes (%) from baseline, mean (SD)	−23.28 (7)	−32.53 (6)	0.380
MCP-1 (pg/mL)			
N	23	22	
Baseline, mean (SD)	124.79 (84.9)	101.47 (47.2)	
End of study, mean (SD)	89.01 (47.4)	79.22 (31.6)	
Changes (%) from baseline, mean (SD)	−28.67 (9)	−21.92 (7)	0.864
hs-CRP (µg/L)			
N	13	11	
Baseline, mean (SD)	1.13 (1.5)	1.14 (1.1)	
End of study, mean (SD)	1.19 (1.5)	0.87 (0.6)	
Changes (%) from baseline, mean (SD)	5.30 (9)	−23.68 (8)	0.282
TNF-α (pg/mL)			
N	13	11	
Baseline, mean (SD)	60.74 (65)	94.23 (123)	
End of study, mean (SD)	19.58 (19.5)	14.94 (5.6)	
Changes (%) from baseline, mean (SD)	−67.76 (12)	−84.14 (10)	0.411

IL: Interleukin; MCP-1: Monocyte Chemoattractant Protein-1; hs-CRP: high sensitivity C-Reactive Protein; TNF-α: Tumor Necrosis Factor-α. Bold in the *p*-value is necessary to stress the significant results.

## Data Availability

The original contributions presented in this study are included in the article. Further inquiries can be directed to the corresponding author.

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
