# Peer review of "Effects of Oral Amino Acid Supplementation on Physical Activity, Systemic Inflammation, and Quality of Life in Adult Patients with Cystic Fibrosis: A Single-Center, Randomized, Double-Blind, Placebo-Controlled Pilot Study"

_nutrients, 2025, doi:10.3390/nu17071239_

Round 1
Reviewer 1 Report
Comments and Suggestions for Authors
Effects of oral amino acid supplementation on physical activity, systemic inflammation and quality of life in adult patients with Cystic Fibrosis: a single center randomized, double blinded, placebo-controlled study.
Nutrients-3507236
Petraglia et al
The authors propose that the study aims to assess the effects of amino acid supplementation on functional status, muscle mass and strength, inflammation, and quality of life in adult CF patients. The authors suggest the study demonstrates that amino acid supplementation improves self-perceived physical performance and that this group exhibited a lower serum IL-6 level compared to the placebo-control group.
Major Reviewer comments:
- Overall study design: The authors should describe why they did not engage subjects to serve as their own controls, i.e. placebo, a wait period then amino-acid supplementation and a cohort with the reverse treatment plan, amino-acid supplementation then a wait period then placebo. This would have provided more robust comparisons of the actual amino acid effect versus placebo.
- Methods: regarding patient recruitment into the 2 arms of the study, the authors should break down the type of mutations CF patents exhibit to demonstrate whether the two groups reflect a somewhat comparative genetic makeup.
- Tables 1and 2: data should reflect the baseline and post placebo or amino acid supplementation periods, this will allow the reader to determine whether changes were noted, if any.
- Results: patient reported perceptions of health differ between the two groups, this data should be showcased, since it is a significant finding reported in the conclusions. Authors are encouraged to demonstrate in a more granular fashion those questions that scored differently.
- Results/discussion: the authors demonstrate that IL-6 serum levels were decreased in the amino acid supplementation group. The authors do not discuss how the data presented is different to that previously described (by some overlapping co-authors) in the 2021 paper demonstrating that physical activity regulates TNF-alpha and IL-6 expression in CF patients (Nigro et al), whereby TNF-alpha levels increased with activity and IL-6 levels did not change. How can the authors be sure the effect on IL-6 is attributed to amino acid or the physical activity tests administered to the patients, and whether the activity level for the patients increased in the amino acid supplementation group overall.
- Methods/Results: The authors should justify why TNF-alpha levels were not measured.
- Conclusion: this is a direct lift of the last paragraph of the discussion. Authors are encouraged to rewrite either one.
Author Response
We sincerely thank the Reviewer for his/her comments. We will try to answer point by point. We are sure that this revision could improve the manuscript and make it more suitable for publication on Nutrients.
- “Overall study design: The authors should describe why they did no engage subjects to serve as their own controls……………. “
We thank the Reviewer for this comment. Although we agree that a cross over study would have provided further evidence of the amino acid positive effects on quality of life, functioning and pro-inflammatory status of patients with Cystic Fibrosis, we want to emphasize that our study design, despite the small sample size, allowed to detect significant differences in terms of health status and systemic inflammation between the two arms. Furthermore, the evident trend of reduction induced by amino-acid supplementation of other pro-inflammatory cytokines, including TNF-alpha that we have now reported in the revised manuscript (Table 5), strongly suggests that the main limitation of the study is more related to the small number of patients enrolled rather than to its double-blind, placebo-controlled design.
- “The authors should break down the type of mutation CF patients exhibit to demonstrate whether the 2 groups reflect a somewhat comparative genetic make-up”
This is an important point and we thank the Reviewer for his/her comment. In the revised manuscript, we have now added a table (Table 3) illustrating the genetic profile of the study population. There were no significant differences in the type of mutation between amino-acid and placebo groups. We have provided additional comments in sections “Results” and “Discussion” (p. 5, lines 198-203 and p.11, lines 349-355, respectively).
- “Tables 2 and 3: data should reflect the baseline and post-placebo or amino acid supplementation period………………….”
We thank the reviewer for this suggestion. In the revised manuscript, we have included a new table (Table 2) presenting the baseline population data, while leaving the table that shows "changes from baseline" unchanged. Table 2 does not include p-values because, from a methodological point of view, it is inappropriate to make statistical inferences at baseline, as the study was not sized to detect differences at enrolment. Inferential tests can be proposed during the statistical analysis phase to explore possible imbalances at baseline, but, according to the CONSORT statement, significance testing of baseline differences in randomized controlled trials should not be performed.
- “Results: patient reported perception of health differ between the two groups, this data should be showcased……………………..”
We thank the Reviewer for his/her comment and agree on the need to better describe the improvement of health perception induced by amino-acid supplementation. In this regard, we have provided to introduce in the revised manuscript a more detailed description of the questionnaires utilized in the present work for the assessment of quality of life in CF patients (Discuss section, p.11, Lines 332-349). This should help the reader to better understand the significance of the single items and to strengthen the importance of these finding especially for a study population characterized, in many cases by poor quality of life.
- “The authors do not discuss how the data presented is different to that previously described by Nigro et al, whereby TNF-alpha levels increased with physical activity and Il-6 levels did not change…..”
We thank the Reviewer for his/her comment. The paper by Nigro et al described as physical activity play a regulation of TNFα and IL-6 expression counteracting inflammation in CF patients. These authors demonstrated that a 3 years regular and moderate physical activity program was associated to a reduction of serum TNF-alpha and low increase or no change in IL-6 levels, thus supporting an overall anti-inflammatory activity induced by exercising (“The present study further examined the relationship between inflammation, physical activity, and clinical phenotype in CF, demonstrating that TNFα levels are statistically lower in active CF patients and negatively correlate with adiponectin while being positively associated with IL-6. The increase in IL-6 following PA has been described as an anti-inflammatory mechanism and a regulator of protein synthesis and body composition”…“PA counteracts inflammation through several molecular mechanisms, including the downregulation of TNFα levels. In line with this evidence, the data analysed in this study show that PA induced a decrease of TNFα in CF patients”.)
The findings of our study do not seem in contrast to how previously reported by literature since they reflect the anti-inflammatory activity of nutritional supplementation which is different to the behavioral intervention represented by a physical activity program. It is likely that exercise and amino acid supplementation may differently regulate the expression of serum TNF-alpha and IL-6.
“How can the authors be sure the effect on IL-6 is attributed to amino acid or the physical activity tests administered to the patients………..”
In this regard, we are profoundly convinced that the observed reduction of IL-6 levels is related to amino acid supplementation. In fact, it is unlikely that a few minutes of physical activity test, administered at baseline and after 4 weeks of treatment, could have affected systemic inflammatory status.
- “The authors should justify why TNF-alpha levels were not measured”
We apologize with the Reviewer for the lack of TNF-alpha measurements in the original version of the manuscript. We have now introduced this data in Table 5. Also, for this cytokine, it was observed a trend to reduction after amino acid supplementation.
- “Conclusion: this is a direct lift of the last paragraph of the Discussion.”
Thank to Reviewer for noticing this mistake. In the revised manuscript, we have provided to eliminate the last paragraph of the discussion, due to a simple duplication error.
Reviewer 2 Report
Comments and Suggestions for Authors
Review of the Manuscript Entitled:
"Effects of Oral Amino Acid Supplementation on Physical Activity, Systemic Inflammation, and Quality of Life in Adult Patients with Cystic Fibrosis: A Single-Center Randomized, Double-Blind, Placebo-Controlled Pilot Study"
Cystic fibrosis is a severe genetic disease, and the search for novel therapeutic strategies to alleviate its symptoms is of great importance. The randomized, double-blind, placebo-controlled trial conducted by the authors presents an interesting approach; however, several aspects should be improved before publication.
- The authors must correct the plagiarized section in lines 182–193, which is directly copied from https://clinicalnutritionespen.com/article/S2405-4577(24)00171-2/fulltext. Self-plagiarism, even from the authors’ previous work, is unethical. This section must be rewritten in original wording to avoid ethical concerns.
- Abstract and Introduction: These sections are well-structured and provide a clear background for the study.
- Methodology: The study design is detailed, and precise. The randomization process and assessment tools are well described.
- Presentation of Results: While the tables provide detailed data, presenting key findings in graphical format would improve readability. Figures can make data interpretation more intuitive for readers. Although not a strict requirement, graphical representation is often preferred in scientific publications. Additionally, Figure 1 should be of higher quality and avoid three-dimensional graphs, which are generally discouraged in scientific literature due to potential misrepresentation of data.
- Some results are reported without in-depth interpretation. For example, the lack of significant differences in physical function tests (e.g., 30-second sit-to-stand, stair climb power test) is noted, but the authors do not provide potential explanations. Addressing possible reasons, such as sample size limitations or test sensitivity, would strengthen the discussion.
- Given that IL-6 was the only inflammatory marker showing a statistically significant reduction, the authors should discuss why other markers (IL-8, IL-10, MCP-1, hsCRP) did not exhibit similar trends. Could the 4-week study duration have been too short to observe broader anti-inflammatory effects?
- As this is a pilot study with a small sample size, the authors should avoid drawing strong conclusions. It would be beneficial to explicitly state that further large-scale studies are required to confirm these findings.
- The manuscript does not mention whether any side effects were observed during the study. Even if no adverse effects occurred, it is important to clearly state this in the discussion.
Generally the language is clear and understandable. Some sentences are too long and complex, making them harder to read. Simplifying them would improve readability (e.g., the long sentence in lines 238–241).
Author Response
We sincerely thank the Reviewer for his/her comments. We will try to answer point by point. We are sure that this revision could improve the manuscript and make it more suitable for publication on Nutrients.
- “The authors must correct the plagiarized section in line 182-193”
We apologize with the Reviewer for this plagiarism, even from the authors’ previous work. In fact, the statistical analysis was performed and written by our statistician, Giorgio Reggiardo, who is the same person appearing as co-author in the manuscript cited by the Reviewer. Nevertheless, in the revised manuscript, we have provided to rewrite the paragraph to avoid ethical concerns.
We sincerely thank the Reviewer for his/her positive comments on Abstract, Introduction and Methods sections.
- “ ….presenting key findings in graphical format would improve readability…….. Figure 1 should be of higher quality and avoid three-dimensional graphs……”
We agree with the Reviewer that presenting findings in graphical format would improve the readability of the results. However, probably due to the great dispersion of values between different cytokines, our attempts to obtain good graphical results were failed. Therefore, we preferred to illustrate our findings only by tables (see Table 5). We have improved the quality of Figure 1 and eliminated three-dimensional graphs.
- “some results are reported without in-depth interpretation…….The lack of significant differences in physical function test is noted, but the authors do not provide potential explanations……
We thank the Reviewer for his/her comment. In the revised manuscript, we integrated more detailed comment in the Discussion section (p. 12, lines 361-371).
- “The authors should discuss why other markers (IL-8, IL-10, MCP-1, hsCRP) did not exhibit similar trends……..”
We thank the Reviewer for his/her comment. Given the strong trend to reduction exhibited by other markers measured in this study (in the revised manuscript, we have also included TNF-alpha), we are strongly convinced that the only reason why these trends did not reach statistically significance was due to small sample size. We already discussed this point in the Discussion section of the original manuscript (p. 11, lines 316-321).
- “………… the authors should avoid drawing strong conclusions……..”
We thank the Reviewer for this comment. We have slightly modified the conclusions. We explained this issue in more detail in the "study limitation" section.
- “The manuscript does not mention whether any side effect were observed during the study……..”
We thank the Reviewer for this suggestion. No side effects were observed during the study in either group and we added this information to line 190 in the results section.
We hope that some changes in "Discuss" section have made the text more fluent and understandable. Tank you for this suggestion.
Round 2
Reviewer 1 Report
Comments and Suggestions for Authors
The authors took on board the reviewer suggestions and generated a much improved manuscript that is more clear and showcases the data generated in the patient-driven study.
Author Response
Comment:
The authors took on board the reviewer suggestions and generated a much improved manuscript that is more clear and showcases the data generated in the patient-driven study.
We thank the Reviewer 1 very much for his suggestions that allowed us to improve the manuscript.